# Universal approximation power of deep residual neural networks via nonlinear control theory

**Paulo Tabuada**
Department of Electrical and Computer Engineering
University of California at Los Angeles,
Los Angeles, CA 90095
`tabuada@ee.ucla.edu`

**Bahman Gharesifard**
Department of Mathematics and Statistics
Queen's University
Kingston, ON, Canada
`bahman.gharesifard@queensu.ca`

## Abstract

In this paper, we explain the universal approximation capabilities of deep residual neural networks through geometric nonlinear control. Inspired by recent work establishing links between residual networks and control systems, we provide a general sufficient condition for a residual network to have the power of universal approximation by asking the activation function, or one of its derivatives, to satisfy a quadratic differential equation. Many activation functions used in practice satisfy this assumption, exactly or approximately, and we show this property to be sufficient for an adequately deep neural network with $n + 1$ neurons per layer to approximate arbitrarily well, on a compact set and with respect to the supremum norm, any continuous function from $\mathbb{R}^n$ to $\mathbb{R}^n$. We further show this result to hold for very simple architectures for which the weights only need to assume two values. The first key technical contribution consists of relating the universal approximation problem to controllability of an ensemble of control systems corresponding to a residual network and to leverage classical Lie algebraic techniques to characterize controllability. The second technical contribution is to identify monotonicity as the bridge between controllability of finite ensembles and uniform approximability on compact sets.

## 1 Introduction

In the past few years, we have witnessed a resurgence in the use of techniques from dynamical and control systems for the analysis of neural networks. This recent development was sparked by the papers (Weinan, 2017; Haber & Ruthotto, 2017; Lu et al., 2018) establishing a connection between certain classes of neural networks, such as residual networks (He et al., 2016), and control systems. However, the use of dynamical and control systems to describe and analyze neural networks goes back at least to the 70's. For example, Wilson-Cowan's equations (Wilson & Cowan, 1972) are differential equations and so is the model proposed by Hopfield in (Hopfield, 1984). These techniques have been used to study several problems such as weight identifiability from data (Albertini & Sontag, 1993; Albertini et al., 1993), controllability (Sontag & Qiao, 1999; Sontag & Sussmann, 1997), and stability (Michel et al., 1989; Hirsch, 1989).

The objective of this paper is to shed new light into the approximation power of deep neural networks and, in particular, of residual deep neural networks (He et al., 2016). It has been empirically observed that deep networks have better approximation capabilities than their shallow counterparts and are easier to train (Ba & Caruana, 2014; Urban et al., 2017). An intuitive explanation for this fact is based on the different ways in which these types of networks perform function approximation. While shallow networks prioritize parallel compositions of simple functions (the number of neurons per layer is a measure of parallelism), deep networks prioritize sequential compositions of simple functions (the number of layers is a measure sequentiality). It is therefore natural to seek insights using control theory where the problem of producing interesting behavior by manipulating a few inputs over time, i.e., by sequentially composing them, has been extensively studied. Even though control-theoretic techniques have been utilized in the literature to showcase the controllabil-

ity properties of neural networks, to the best of our knowledge, this paper is the first to use tools from geometric control theory to establish universal approximation properties with respect to the infinity norm.

## 1.1 CONTRIBUTIONS

In this paper we focus on residual networks (He et al., 2016). This being said, as explained in (Lu et al., 2018), similar techniques can be exploited to analyze other classes of networks. It is known that deep residual networks have the power of universal approximation. What is less understood is where this power comes from. We show in this paper that it stems from the activation functions in the sense that when using a sufficiently rich activation function, even networks with very simple architectures and weights taking only two values suffice for universal approximation. It is the power of sequential composition, analyzed in this paper via geometric control theory, that unpacks the richness of the activation function into universal approximability. Surprisingly, the level of richness required from an activation function also has a very simple characterization; it suffices for activation functions (or a suitable derivative) to satisfy a quadratic differential equation. Most activation functions in the literature either satisfy this condition or can be suitably approximated by functions satisfying it.

More specifically, given a finite ensemble of data points, we cast the problem of designing weights for training a deep residual network as the problem of driving the state of a finite ensemble of initial points with a single open-loop control input to the finite ensemble of target points produced by the function to be learned when evaluated at the initial points. In spite of the fact that we only have access to a single open-loop control input, we prove that the corresponding ensemble of control systems is controllable. This result can also be understood in terms of the memorization capacity of deep networks, almost any finite set of samples can be memorized, see (Yun et al., 2019; Vershynin, 2020) for some recent work on this problem. We then utilize this controllability property to obtain universal approximability results for continuous functions in a uniform sense, i.e., with respect to the supremum norm. This is achieved by using the notion of monotonicity that lets us conclude uniform approximability on compact sets from controllability of finite ensembles.

## 1.2 RELATED WORK

Several papers have studied and established that residual networks have the power of universal approximation. This was done in (Lin & Jegelka, 2018) by focusing on the particular case of residual networks with the ReLU activation function. It was shown that any such network with $n$ states and one neuron per layer can approximate an arbitrary Lebesgue integrable function $f : \mathbb{R}^n \to \mathbb{R}$ with respect to the $L^1$ norm. The paper (Zhang et al., 2019) shows that the functions described by deep networks with $n$ states per layer, when these networks are modeled as control systems, are restricted to be homeomorphisms. The authors then show that increasing the number of states per layer to $2n$ suffices to approximate arbitrary homeomorphisms $f : \mathbb{R}^n \to \mathbb{R}^n$ under the assumption the underlying network already has the power of universal approximation. Note that the results in (Lin & Jegelka, 2018) do not model deep networks as control systems and, for this reason, bypass the homeomorphism restriction. There is also an important distinction to be made between requiring a network to exactly implement a function and to approximate it. The homeomorphism restriction does not prevent a network from approximating arbitrary functions; it just restricts the functions that can be implemented as a network. Closer to this paper are the results in (Li et al., 2019) establishing universal approximation, with respect to the $L^p$ norm, $1 \le p < \infty$, based on a general sufficient condition satisfied by several examples of activation functions. These results are a major step forward in identifying what is needed for universal approximability, as they are not tied to specific architectures or activation functions. In this paper we establish universal approximation in the stronger sense of the infinity norm $L^\infty$ which implies, as a special case, universal approximation with respect to the $L^p$ norm for $1 \le p < \infty$.

At the technical level, our results build upon the controllability properties of deep residual networks. Earlier work on controllability of differential equation models for neural networks, e.g., (Sontag & Qiao, 1999), assumed the weights to be constant and that an exogenous control signal was fed into the neurons. In contrast, we regard the weights as control inputs and that no additional control inputs are present. These two different interpretations of the model lead to two very different

technical problems. More recent work in the control community includes (Agrachev & Caponigro, 2009), where it is shown that any orientation preserving diffeomorphism on a compact manifold, can be obtained as the flow of a control system when using a time-varying feedback controller. In the context of this paper those results can be understood as: residual networks can represent any orientation preserving diffeomorphism provided that we can make the weights depend on the state. Although quite insightful, such results are not applicable to the standard neural network models where the weights are not allowed to depend on the state. Another relevant topic is ensemble control. Most of the work on the control of ensembles, see for instance (Li & Khaneja, 2006; Helmke & Schönlein, 2014; Brockett, 2007), considers parametrized ensembles of vector fields. In other words, the individual systems that drive the state of the whole ensemble are different, whereas in our setting the ensemble consists of exact copies of the same system, albeit initialized differently. In this sense, our work is most closely related to the setting of (Agrachev & Sarychev, 2020a;b) where controllability results for ensembles of infinitely many control systems are provided. In this paper, in contrast, we use Lie algebraic techniques to study controllability of finite ensembles and obtain approximation results for infinite ensembles by using the notion of monotonicity rather than Lie algebraic techniques as is done in (Agrachev & Sarychev, 2020a;b). Moreover, by focusing on the specific control systems arising from deep residual networks we are able to provide easier to verify controllability conditions than those provided in (Agrachev & Sarychev, 2020a;b) for more general control systems. Controllability of finite ensembles of control systems motivated by neural network applications was investigated in (Cuchiero et al., 2019) where it is shown that controllability is a generic property and that, for control systems that are linear in the inputs, 5 inputs suffice. These results are insightful but they do not apply to specific control systems such as those describing residual networks and studied in this paper. Moreover the results in (Cuchiero et al., 2019) do not address the problem of universal approximation in the infinity norm.

To conclude the review of related work, we note that universal approximation with respect to the infinity norm for non-residual deep networks, allowing for general classes of activation functions, was recently established in (Kidger & Lyons, 2020). In particular, it is shown in (Kidger & Lyons, 2020) that under very mild conditions on the activation functions any continuous function $f : K \to \mathbb{R}^m$, where $K \subset \mathbb{R}^n$ is compact, can be approximated in the infinity norm using a deep neural network of width $n + m + 2$. Even though these results do not directly apply to residual networks (due to the presence of skip connections), they require $2n + 2$ neurons for the case discussed in this paper where $n = m$. In contrast, one of our main results, Corollary 4.5, asserts that a width of $n + 1$ is sufficient for universal approximation.

When preparing the final version of this paper we became aware of the paper (Park et al., 2021), published at the same venue and addressing a similar problem. Although the results of (Park et al., 2021) do not directly apply to residual networks (due to the presence of skip connections), they establish that $\max\{n + 1, m\}$ neurons per layer suffice for universal approximation. When $n = m$, the case discussed in this paper, both papers provide the same number of neurons $n + 1$. However, there are several technical nuances across the results (for instance, some results in (Park et al., 2021) require a layer of step activation functions, in addition to, e.g., ReLUs, whereas others hold even if the domain of $f$ is not compact when approximating in the $L^p$ norm) that deserve further investigation.

## 2 CONTROL-THEORETIC VIEW OF RESIDUAL NETWORKS

### 2.1 FROM RESIDUAL NETWORKS TO CONTROL SYSTEMS AND BACK

We start by providing a control system perspective on residual neural networks. We mostly follow the treatment proposed in (Weinan, 2017; Haber & Ruthotto, 2017; Lu et al., 2018), where it was suggested that residual neural networks with an update equation of the form:

$$x(k + 1) = x(k) + S(k)\Sigma(W(k)x(k) + b(k)), \tag{2.1}$$

where $k \in \mathbb{N}_0$ indexes each layer, $x(k) \in \mathbb{R}^n$, and $(S(k), W(k), b(k)) \in \mathbb{R}^{n \times n} \times \mathbb{R}^{n \times n} \times \mathbb{R}^n$, can be interpreted as a control system when $k$ is viewed as indexing time. In (2.1), $S$, $W$, and $b$ are the weights functions assigning weights to each time instant $k$, and $\Sigma : \mathbb{R}^n \to \mathbb{R}^n$ is of the form $\Sigma(x) = (\sigma(x_1), \sigma(x_2), \dots, \sigma(x_n))$, where $\sigma : \mathbb{R}^n \to \mathbb{R}^n$ is an *activation function*. By drawing an analogy between (2.1) and Euler's forward method to discretize differential equations, one can

interpret (2.1) as the time discretization of the continuous-time control system:

$$\dot{x}(t) = S(t)\Sigma(W(t)x(t) + b(t)), \qquad (2.2)$$

where $x(t) \in \mathbb{R}^n$ and $(S(t), W(t), b(t)) \in \mathbb{R}^{n \times n} \times \mathbb{R}^{n \times n} \times \mathbb{R}^n$; in what follows, and in order to make the presentation simpler, we sometimes drop the dependency on time. To make the connection between the discretization and (2.2) precise, let $x : [0, \tau] \to \mathbb{R}^n$ be a solution of the control system (2.2) for the control input $(S, W, b) : [0, \tau] \to \mathbb{R}^{n \times n} \times \mathbb{R}^{n \times n} \times \mathbb{R}^n$, where $\tau \in \mathbb{R}^+$. Then, given any desired accuracy $\varepsilon \in \mathbb{R}^+$ and any norm $|\cdot|$ in $\mathbb{R}^n$, there exists a sufficiently small time step $T \in \mathbb{R}^+$ so that the function $z : \{0, 1, \dots, \lfloor \tau/T \rfloor\} \to \mathbb{R}^n$ defined by:

$$z(0) = x(0), \quad z(k+1) = z(k) + TS(kT)\Sigma(W(kT)z(k) + b(kT)),$$

approximates the sequence $\{x(kT)\}_{k=0,\dots,\lfloor \tau/T \rfloor}$ with error $\varepsilon$, i.e.:

$$|z(k) - x(kT)| \leq \varepsilon,$$

for all $k \in \{0, 1, \dots, \lfloor \tau/T \rfloor\}$. Intuitively, any statement about the solutions of (2.2) holds for the solutions of (2.1) with arbitrarily small error $\varepsilon$, provided that we can choose the depth to be arbitrarily large since by making $T$ small we increase the depth, given by $1 + \lfloor \tau/T \rfloor$.

## 2.2 NEURAL NETWORK TRAINING AND CONTROLLABILITY

Given a function $f : \mathbb{R}^n \to \mathbb{R}^n$ and a finite set of samples $E_{\text{samples}} \subset \mathbb{R}^n$, the problem of training a residual network so that it maps $x \in E_{\text{samples}}$ to $f(x)$ can be phrased as the problem of constructing an open-loop control input $(S, W, b) : [0, \tau] \to \mathbb{R}^{n \times n} \times \mathbb{R}^{n \times n} \times \mathbb{R}^n$ so that the resulting solution of (2.2) takes the states $x \in E_{\text{samples}}$ to the states $f(x)$. It should then come as no surprise that the ability to approximate a function $f$ is tightly connected with the control-theoretic problem of controllability: given, one initial state $x^{\text{init}} \in \mathbb{R}^n$ and one final state $x^{\text{fin}} \in \mathbb{R}^n$, when does there exist a finite time $\tau \in \mathbb{R}^+$ and a control input $(S, W, b) : [0, \tau] \to \mathbb{R}^{n \times n} \times \mathbb{R}^{n \times n} \times \mathbb{R}^n$ so that the solution of (2.2) starting at $x^{\text{init}}$ at time 0 ends at $x^{\text{fin}}$ at time $\tau$?

To make the connection between controllability and the problem of mapping every $x \in E_{\text{samples}}$ to $f(x)$ clear, it is convenient to consider the ensemble of $d = |E_{\text{samples}}|$ copies of (2.2) given by the matrix differential equation:

$$\dot{X}(t) = [S(t)\Sigma(W(t)X_{\bullet 1}(t) + b(t))|S(t)\Sigma(W(t)X_{\bullet 2}(t) + b(t))| \dots |S(t)\Sigma(WX_{\bullet d}(t) + b(t)))], \tag{2.3}$$

where for time $t \in \mathbb{R}_0^+$ the $i$th column of the matrix $X(t) \in \mathbb{R}^{n \times d}$, denoted by $X_{\bullet i}(t)$, is the solution of the $i$th copy of (2.2) in the ensemble. If we now index the elements of $E_{\text{samples}}$ as $\{x^1, \dots, x^d\}$, where $d$ is the cardinality of $E_{\text{samples}}$, and consider the matrices $X^{\text{init}} = [x^1|x^2| \dots |x^d]$ and $X^{\text{fin}} = [f(x^1)|f(x^2)| \dots |f(x^d)]$, we see that the existence of a control input resulting in a solution of (2.3) starting at $X^{\text{init}}$ and ending at $X^{\text{fin}}$, i.e., controllability of (2.3), is equivalent to existence of an input for (2.2) so that the resulting solution starting at $x^i \in E_{\text{samples}}$ ends at $f(x^i)$, for all $i \in \{1, \dots, d\}$.

Note that achieving controllability of (2.3) is especially difficult, since all the copies of (2.2) in (2.3) are *identical* and they all use the *same input*. Therefore, to achieve controllability, we must have sufficient diversity in the initial conditions to overcome the symmetries present in (2.3), see (Aguilar & Gharesifard, 2014). Our controllability result, Theorem 4.2, describes precisely such diversity. As mentioned in the introduction, this observation also distinguishes the problem under study here from the classical setting of ensemble control (Li & Khaneja, 2006; Helmke & Schönlein, 2014), with the exception of the recent work (Cuchiero et al., 2019; Agrachev & Sarychev, 2020a;b), where a collection of systems with *different* dynamics are driven by the same control input.

## 3 PROBLEM FORMULATION

Our starting point is the control system:

$$\dot{x}(t) = s(t)\Sigma(W(t)x(t) + b(t)), \qquad (3.1)$$

a slightly simplified version of (2.2), where $x(t) \in \mathbb{R}^n$, $(s(t), W(t), b(t)) \in \mathbb{R} \times \mathbb{R}^{n \times n} \times \mathbb{R}^n$, and the input $S$ in (2.2) is now the scalar-valued function $s$; as we will prove in what follows, this model

is enough for universal approximation. In fact, we will later see[1] that it suffices to let $s$ assume two arbitrary values only (one positive and one negative). Moreover, for certain activation functions, we can dispense with $s$ altogether.

**Assumption 1.** *We make the following assumptions regarding the model* (3.1)*:*

- *The function $\Sigma$ is defined as $\Sigma : x \mapsto (\sigma(x_1), \sigma(x_2), \ldots, \sigma(x_n))$, where the activation function $\sigma : \mathbb{R} \to \mathbb{R}$, or a suitable derivative of it, satisfies a quadratic differential equation, i.e., $D\xi = a_0 + a_1 \xi + a_2 \xi^2$ with $a_1, a_2, a_3 \in \mathbb{R}$, $a_2 \neq 0$, and $\xi = D^j \sigma$ for some $j \in \mathbb{N}_0$. Here, $D^j \sigma$ denotes the derivative of $\sigma$ of order $j$ and $D^0 \sigma = \sigma$.*

- *The activation function $\sigma : \mathbb{R} \to \mathbb{R}$ is Lipschitz continuous, $D\sigma \geq 0$, and $\xi = D^j \sigma$ defined above is injective.*

Table 1: Activation functions and the differential equations they satisfy.

| Function name | Definition | Satisfied differential equation |
|---|---|---|
| Logistic function | $\sigma(x) = \frac{1}{1+e^{-x}}$ | $D\sigma - \sigma + \sigma^2 = 0$ |
| Hyperbolic tangent | $\sigma(x) = \frac{e^x - e^{-x}}{e^x + e^{-x}}$ | $D\sigma - 1 + \sigma^2 = 0$ |
| Soft plus | $\sigma(x) = \frac{1}{r} \log(1 + e^{rx})$ | $D^2\sigma - rD\sigma + r(D\sigma)^2 = 0$ |

Several activation functions used in the literature are solutions of quadratic differential equations as can be seen in Table 1. Moreover, activation functions that are not differentiable can also be handled via approximation. For example, the ReLU function defined by $\max\{0, x\}$ can be approximated by $\sigma(x) = \log(1 + e^{rx})/r$, as $r \to \infty$, which satisfies the quadratic differential equation given in Table 1. Similarly, the leaky ReLU, defined by $\sigma(x) = x$ for $x \geq 0$ and $\sigma(x) = rx$ for $x < 0$, is the limit as $k \to \infty$ of $\alpha(x) = rx + \log(1 + e^{(1-r)kx})/k$, and the function $\alpha$ satisfies $D^2\alpha - k(1+r)D\alpha + k(D\alpha)^2 + kr = 0$.

The Lipschitz continuity assumption is made to simplify the presentation and can be replaced with local Lipschitz continuity, which then does not need to be assumed, since $\sigma$ is analytic in virtue of being the solution of an analytic (quadratic) differential equation. Moreover, all the activation functions in Table 1 are Lipschitz continuous, have positive derivative and are thus injective.

To formally state the problem under study in this paper, we need to discuss a different point of view on the solutions of the control system (3.1) given by *flows*. A continuously differentiable curve $x : [0, \tau] \to \mathbb{R}^n$ is said to be a solution of (3.1) under the piecewise continuous input $(s, W, b) : [0, \tau] \to \mathbb{R} \times \mathbb{R}^{n \times n} \times \mathbb{R}^n$ if it satisfies (3.1). Under the stated assumptions on $\sigma$, given a piecewise continuous input and a state $x^{\text{init}} \in \mathbb{R}^n$, there is one and at most one solution $x(t)$ of (3.1) satisfying $x(0) = x^{\text{init}}$. Moreover, solutions are defined for all $\tau \in \mathbb{R}_0^+$. We can thus define the flow of (3.1) under the input $(s, W, b)$ as the map $\phi^\tau : \mathbb{R}^n \to \mathbb{R}^n$ given by the assignment $x^{\text{init}} \mapsto x(\tau)$. In other words, $\phi^\tau(x^{\text{init}})$ is the point reached at time $\tau$ by the unique solution starting at $x^{\text{init}}$ at time $0$. When the time $\tau$ is clear from context, we denote a flow simply by $\phi$. It will also be convenient to denote the flow $\phi^\tau$ by $Z^\tau$ when $\phi$ is defined by the solution of the differential equation $\dot{x} = Z(x)$ for some vector field $Z : \mathbb{R}^n \to \mathbb{R}^n$.

We will use flows to approximate arbitrary continuous functions $f : \mathbb{R}^n \to \mathbb{R}^m$. Since flows have the same domain and co-domain, and $f : \mathbb{R}^n \to \mathbb{R}^m$ may not, we first lift $f$ to a map $\tilde{f} : \mathbb{R}^k \to \mathbb{R}^k$. When $n > m$, we lift $f$ to $\tilde{f} = \imath \circ f : \mathbb{R}^n \to \mathbb{R}^n$, where $\imath : \mathbb{R}^m \to \mathbb{R}^n$ is the injection given by $\imath(x) = (x_1, \ldots, x_n, 0, \ldots, 0)$. In this case $k = n$. When $n < m$, we lift $f$ to $\tilde{f} = f \circ \pi : \mathbb{R}^m \to \mathbb{R}^m$, where $\pi : \mathbb{R}^m \to \mathbb{R}^n$ is the projection $\pi(x_1, \ldots, x_n, x_{n+1}, \ldots, x_m) = (x_1, \ldots, x_n)$. In this case $k = m$. Although we could consider factoring $f$ through a map $g : \mathbb{R}^n \to \mathbb{R}^m$, i.e., to construct $\tilde{f} : \mathbb{R}^n \to \mathbb{R}^n$ so that $f = g \circ \tilde{f}$ as done in, e.g., (Li et al., 2019), the construction of $g$ requires a deep understanding of $f$, since a necessary condition for this factorization is $f(\mathbb{R}^n) \subseteq g(\mathbb{R}^n)$. Constructing $g$ so as to contain $f(\mathbb{R}^n)$ on its image requires understanding what $f(\mathbb{R}^n)$ is and this information is not available in learning problems. Given this discussion, in the remainder of this paper we directly assume we seek to approximate a map $f : \mathbb{R}^n \to \mathbb{R}^n$.

---

[1] See the discussion after the proof of Theorem 4.2 in (Tabuada & Gharesifard, 2020).

The final ingredient we need before stating the problem solved in this paper is the precise notion of approximation. Throughout the paper, we will investigate approximation in the sense of the $L^\infty$ (supremum) norm, i.e.:

$$\|f\|_{L^\infty(E)} = \sup_{x \in E} |f(x)|_\infty,$$

where $E \subset \mathbb{R}^n$ is the compact set over which is the approximation is going to be conducted and $|f(x)|_\infty = \max_{i \in \{1,\dots,n\}} |f_i(x)|$. Some approximation results will be stated for networks modeled by a control system (3.1) with state space $\mathbb{R}^n$. In such cases the approximation quality is measured by $\|f - \phi\|_{L^\infty(E)}$ where $\phi$ is the flow of (3.1). Other results will require networks with state space $\mathbb{R}^m$ for $m > n$. For those cases the approximation quality is measured by $\|f - \beta \circ \phi \circ \alpha\|_{L^\infty(E)}$ where $\alpha : \mathbb{R}^n \to \mathbb{R}^m$ is an injection and $\beta : \mathbb{R}^m \to \mathbb{R}^n$ is a projection. These maps will be linear and can be implemented as the first and last layers of a residual network.

We are now ready to state the two problems we study in this paper.

**Problem 3.1.** *Let $f : \mathbb{R}^n \to \mathbb{R}^n$ be a continuous function, $E_{\text{samples}} \subset \mathbb{R}^n$ be a finite set, and $\varepsilon \in \mathbb{R}_0^+$ be the desired approximation accuracy. Under Assumption 1, does there exist a time $\tau \in \mathbb{R}^+$ and an input $(s, W, b) : [0, \tau] \to \mathbb{R} \times \mathbb{R}^{n \times n} \times \mathbb{R}^n$ so that the flow $\phi^\tau : \mathbb{R}^n \to \mathbb{R}^n$ defined by the solution of (3.1) with state space $\mathbb{R}^n$ under the said input satisfies:*

$$\|f - \phi^\tau\|_{L^\infty(E_{\text{samples}})} \le \varepsilon?$$

Note that we allow $\varepsilon$ to be zero in which case the flow $\phi^\tau$ matches $f$ exactly on $E_{\text{samples}}$, i.e., $f(x) = \phi^\tau(x)$ for every $x \in E_{\text{samples}}$. The next problem considers the more challenging case of approximation on compact sets and allows for residual networks with $m > n$ neurons per layer when approximating functions on $\mathbb{R}^n$.

**Problem 3.2.** *Let $f : \mathbb{R}^n \to \mathbb{R}^n$ be a continuous function, $E \subset \mathbb{R}^n$ be a compact set, and $\varepsilon \in \mathbb{R}^+$ be the desired approximation accuracy. Under Assumption 1, does there exist $m \in \mathbb{N}$, a time $\tau \in \mathbb{R}^+$, an injection $\alpha : \mathbb{R}^n \to \mathbb{R}^m$, a projection $\beta : \mathbb{R}^m \to \mathbb{R}^n$, and an input $(s, W, b) : [0, \tau] \to \mathbb{R} \times \mathbb{R}^{n \times n} \times \mathbb{R}^n$ such that the flow $\phi^\tau : \mathbb{R}^n \to \mathbb{R}^n$ defined by the solution of (3.1) with state space $\mathbb{R}^{n+1}$ under the said input satisfies:*

$$\|f - \beta \circ \phi^\tau \circ \alpha\|_{L^\infty(E)} \le \varepsilon?$$

In the next section, we will show the answer to both these problems to be affirmative. For the second problem we can take $m = n$ under the additional assumption of monotonicity which is satisfied by construction when we allow $m = n + 1$ neurons per layer.

# 4 MAIN RESULTS

The proofs of all the results in this section are provided in (Tabuada & Gharesifard, 2020).

We first discuss the problem of constructing an input for (3.1) so that the resulting flow $\phi$ satisfies $\phi(x) = f(x)$ for all the points $x$ in a given finite set $E_{\text{samples}} \subset \mathbb{R}^n$. We explained in Section 2.2 that this is equivalent to determining if the ensemble control system (2.3) is controllable. It is simple to see that controllability of (2.3) cannot hold on all of $\mathbb{R}^{n \times d}$, since if the initial state $X(0)$ satisfies $X_{\bullet i}(0) = X_{\bullet j}(0)$ for some $i \ne j$, we must have $X_{\bullet i}(t) = X_{\bullet j}(t)$ for all $t \in [0, \tau]$ by uniqueness of solutions of differential equations.

Our first result establishes that the controllability property holds for the ensemble control system (2.3) on a dense and connected submanifold of $\mathbb{R}^{n \times d}$, independently of the (finite) number of copies $d$, as long the activation function satisfies Assumption 1. Before stating this result, we recall the formal definition of controllability.

**Definition 4.1.** *A point $X^{\text{fin}} \in \mathbb{R}^{n \times d}$ is said to be reachable from a point $X^{\text{init}} \in \mathbb{R}^{n \times d}$ for the control system (2.3) if there exist $\tau \in \mathbb{R}^+$ and a control input $(s, W, b) : [0, \tau] \to \mathbb{R} \times \mathbb{R}^{n \times n} \times \mathbb{R}^n$ so that the solution $X$ of (2.3) under said input satisfies $X(0) = X^{\text{init}}$ and $X(\tau) = X^{\text{fin}}$. Control system (2.3) is said to be controllable on a submanifold $M$ of $\mathbb{R}^{n \times d}$ if any point in $M$ is reachable from any point in $M$.*

**Theorem 4.2.** *Let $N \subset \mathbb{R}^{n \times d}$ be the set defined by:*

$$N = \left\{ A \in \mathbb{R}^{n \times d} \mid \prod_{1 \leq i < j \leq d} (A_{\ell i} - A_{\ell j}) = 0, \; \ell \in \{1, \ldots, n\} \right\}.$$

*Let $n > 1$ and suppose that Assumption 1 holds. Then the ensemble control system* (2.3) *is controllable on the submanifold $M = \mathbb{R}^{n \times d} \backslash N$.*

It is worth mentioning that the assumption of $n \neq 1$ ensures connectedness of the submanifold $M$, which we rely on to obtain controllability. The following corollary of Theorem 4.2 weakens controllability to reachability but applies to a larger set.

**Corollary 4.3.** *Let $M \subset \mathbb{R}^{n \times d}$ be the submanifold defined in Theorem 4.2. Under assumptions of Theorem 4.2, any point in $M$ is reachable from a point $A \in \mathbb{R}^{n \times d}$ for which:*

$$A_{\bullet i} \neq A_{\bullet j},$$

*holds for all $i \neq j$, where $i, j \in \{1, \ldots, d\}$.*

The assumption $A_{\bullet i} \neq A_{\bullet j}$ in Corollary 4.3 requires all the columns of $A$ to be different and is always satisfied when $A = \begin{bmatrix} x^1 | x^2 | \ldots | x^d \end{bmatrix}$, $x^i \in E_{\text{samples}}$. Hence, for any finite set $E_{\text{samples}}$ there exists a flow $\phi$ of (3.1) satisfying $f(x) = \phi(x)$ for all $x \in E_{\text{samples}}$ provided that $f(E_{\text{samples}}) \subset M$, i.e., Problem 3.1 is solved with $\varepsilon = 0$. Moreover, since $M$ is dense in $\mathbb{R}^{n \times d}$, when $f(E_{\text{samples}}) \subset M$ fails, there still exists a flow $\phi$ of (3.1) taking $\phi(x)$ arbitrarily close to $f(x)$ for all $x \in E_{\text{samples}}$, i.e., Problem 3.1 is solved for any $\varepsilon > 0$. This result also sheds light on the memorization capacity of residual networks as it states that almost any finite set of samples can be memorized, independently of its cardinality. See, e.g., (Yun et al., 2019; Vershynin, 2020), for recent results on this problem that do not rely on differential equation models.

Some further remarks are in order. Theorem 4.2 and Corollary 4.3 do not directly apply to the ReLU activation function, defined by $\max\{0, x\}$, since this function is not differentiable. However, the ReLU is approximated by the activation function:

$$\frac{1}{r} \log(1 + e^{rx}),$$

as $r \to \infty$. In particular, as $r \to \infty$ the ensemble control system (2.3) with $\sigma(x) = \log(1 + e^{rx})/r$ converges to the ensemble control system (2.3) with $\sigma(x) = \max\{0, x\}$ and thus the solutions of the latter are arbitrarily close to the solutions of the former whenever $r$ is large enough. Moreover, $\xi = D\sigma$ satisfies $D\xi = r\xi - r\xi^2$ and $D\xi = re^{rx}/(1 + e^{rx})^2 > 0$ for $x \in \mathbb{R}$ and $r > 0$ thus showing that $\xi$ is an increasing function and, consequently, injective.

The conclusions of Theorem 4.2 and Corollary 4.3 also hold if we weaken the assumptions on the inputs of (3.1). It suffices for the entries of $W$ and $b$ to take values on a set with two elements (one positive and one negative), see the discussion after the proof of Theorem 4.2 in (Tabuada & Gharesifard, 2020) for details. Moreover, when the activation function is and odd function, i.e., $\sigma(-x) = -\sigma(x)$, as is the case for the hyperbolic tangent, the conclusions of Theorem 4.2 hold for the simpler version of (3.1), where we fix $s$ to be 1.

In order to extend the approximation guarantees from a finite set $E_{\text{samples}} \subset \mathbb{R}^n$ to an arbitrary compact set $E \subset \mathbb{R}^n$, we rely on the notion of monotonicity. On $\mathbb{R}^n$ we consider the ordering relation $x \preceq x'$ defined by $x_i \leq x_i'$ for all $i \in \{1, \ldots, n\}$ and $x, x' \in \mathbb{R}^n$. A map $f : \mathbb{R}^n \to \mathbb{R}^n$ is said to be monotone when it respects this ordering relation, i.e., when $x \preceq x'$ implies $f(x) \preceq f(x')$. When $f$ is continuous differentiable, monotonicity admits a simple characterization (Hirsch & Smith, 2006):

$$\frac{\partial f_i}{\partial x_j} \geq 0, \quad \forall i, j \in \{1, \ldots, n\}. \tag{4.1}$$

A vector field $Z : \mathbb{R}^n \to \mathbb{R}^n$ is said to be monotone when its flow $\phi^\tau : \mathbb{R}^n \to \mathbb{R}^n$ is a monotone map. Monotone vector fields admit a characterization similar to (4.1), see (Smith, 2008):

$$\frac{\partial Z_i}{\partial x_j} \geq 0, \quad \forall i, j \in \{1, \ldots, n\}, i \neq j. \tag{4.2}$$

**Theorem 4.4.** *Let $n > 1$ and suppose that Assumption 1 holds. Then, for every monotone analytic function $f : \mathbb{R}^n \to \mathbb{R}^n$, for every compact set $E \subset \mathbb{R}^n$, and for every $\varepsilon \in \mathbb{R}^+$ there exist a time $\tau \in \mathbb{R}^+$ and an input $(s, W, b) : [0, \tau] \to \mathbb{R} \times \mathbb{R}^{n \times n} \times \mathbb{R}^n$ so that the flow $\phi^\tau : \mathbb{R}^n \to \mathbb{R}^n$ defined by the solution of* (3.1) *with state space $\mathbb{R}^n$ under the said input satisfies:*

$$\|f - \phi^\tau\|_{L^\infty(E)} \leq \varepsilon. \tag{4.3}$$

When the function to be approximated is simply continuous, we can first approximate it by a polynomial function using Stone-Weierstass' Theorem and then embed it into a monotone function. Although such embeddings typically require doubling the dimension, we leverage the existence of the projection map $\beta$ to introduce a novel embedding that only requires increasing $n$ to $n + 1$.

**Corollary 4.5.** *Let $n > 1$ and suppose that Assumption 1 holds. Then, for every continuous function $f : \mathbb{R}^n \to \mathbb{R}^n$, for every compact set $E \subset \mathbb{R}^n$, and for every $\varepsilon \in \mathbb{R}^+$ there exist a time $\tau \in \mathbb{R}^+$, an injection $\alpha : \mathbb{R}^n \to \mathbb{R}^{n+1}$, a projection $\beta : \mathbb{R}^{n+1} \to \mathbb{R}^n$, and an input $(s, W, b) : [0, \tau] \to \mathbb{R} \times \mathbb{R}^{(n+1) \times (n+1)} \times \mathbb{R}^{n+1}$ so that the flow $\phi^\tau : \mathbb{R}^{n+1} \to \mathbb{R}^{n+1}$ defined by the solution of* (3.1) *with state space $\mathbb{R}^{n+1}$ under the said input satisfies:*

$$\|f - \beta \circ \phi^\tau \circ \alpha\|_{L^\infty(E)} \leq \varepsilon.$$

It is worth pointing out that, contrary to Theorem 4.4, no requirements are placed on $f$ in addition to continuity. In (Agrachev & Sarychev, 2020a;b), sufficient conditions for the existence of a flow $\phi^\tau$ satisfying (4.3) are given for a more general class of control systems. The assumptions used in Theorem (4.4) are not easy to compare with the assumptions in Theorem 5.1 of (Agrachev & Sarychev, 2020b). However, by employing a deep network of width $n + 1$, i.e., by using Corollary 4.5, we rely on much simpler assumptions on the activation functions which are satisfied by the networks used in practice. In contrast, (Agrachev & Sarychev, 2020b, Theorem 5.1) requires a strong Lie algebra approximation property to be satisfied by the ensemble control system that does not appear to be easy to verify.

KEY NOVEL TECHNICAL IDEAS

We find it fruitful to sketch the proofs, available in (Tabuada & Gharesifard, 2020), of the main results to give the reader a flavor of some of the novel technical ideas.

- In Theorem 4.2, we use the notion of ensemble controllability to find the open-loop control inputs that map the finite set of sample $E_{\text{samples}} \subset \mathbb{R}^n$ to the corresponding state under $f$. The use of Lie algebraic techniques to study controllability is standard practice in nonlinear control (Jurdjevic, 1996). To give the unfamiliar reader a glimpse on this, given two smooth vector fields $f_1, f_2 : \mathbb{R}^n \to \mathbb{R}^n$, one can show that the reachable set of the control system $\dot{x} = u_1 f_1(x) + u_2 f_2(x)$, i.e., the set of points that can be reached by selecting $u_1$ and $u_2$ appropriately, is equivalent to the one for $\dot{x} = u_1 f_1(x) + u_2 f_2(x) + u_3 [f_1, f_2](x)$, where $[f_1, f_2](x) := \frac{\partial f_2}{\partial x} f_1(x) - \frac{\partial f_1}{\partial x} f_2(x)$ is the Lie bracket of $f_1$ and $f_2$. Continuing in this fashion, the closure of the set of all brackets generated by $f_1$ and $f_2$, i.e., the Lie algebra generated by these, corresponds to the directions that we can steer the system to. Now the setting considered here is far from this classical setting and three challenges arise when attempting to apply these techniques to (2.3). The first one is that we are steering an ensemble of data points, rather than one point, using a single control system. The second challenge is that most existing Lie algebraic controllability results are for control systems that are affine in the input, such as the one above, whereas (2.3) is not. To address this challenge, we identified a subset of inputs that, when used in a piecewise constant manner, lead to a control system that is easier to analyze. The third challenge is that computing the dimension of the associated Lie algebra is difficult, since one has to establish linear (in)dependence of highly nonlinear functions. To proceed, we had to identify which assumptions to place on $\sigma$ so that the dimension of the Lie algebra could be computed while allowing for a quite general class of activation functions. Strikingly, by assuming that $\sigma$ satisfies a quadratic differential equation, which still includes many of the known activation functions, we were able to provide a closed form expression for the determinant of the matrix (A.1) in Lemma A.1 in (Tabuada & Gharesifard, 2020) thereby showing that, for every point in the open and dense submanifold $M$, the dimension of the Lie algebra is the dimension of the state space.

- The novel idea in the proof of Theorem 4.4 is the link between monotonicity and approximability in the $L^\infty$ sense. We already know from Theorem 4.2 that deep residual networks can memorize to arbitrary accuracy any continuous function $f$ evaluated on a finite sample set $E_{\text{samples}}$. The question left open is, what is the mismatch between $f$ and the flow of (3.1) for points in the domain of $f$ but not in $E_{\text{samples}}$? Lemma A.3 in (Tabuada & Gharesifard, 2020) provides an upper bound for such gap provided that the flow of (3.1) is monotone. With this upper bound in hand, the proof of Theorem 4.2 shows that when $f$ is monotone, there exists a monotone flow of (3.1) that approximates, with arbitrary accuracy, $f$ evaluated on $E_{\text{samples}}$ and the result then follows. In summary, the proof of Theorem 4.4 establishes that monotonocity is the key property that allows to extend approximation on finite sample sets (Theorem 4.2) to approximation on compact sets (Theorem 4.4).

- Our final step is in Corollary 4.5 where we construct an embedding of $f$ into a monotone mapping by increasing the domain and co-domain of $f$ by one, i.e., by working on $\mathbb{R}^{n+1}$. Embedding non-monotone maps into monotone maps typically requires doubling the dimension. The key novel idea here is to add a linear correction to $f$ that is compensated by the linear function $\beta$ implemented as the last layer of a deep residual network. With this linear compensation it suffices to increase the domain and co-domain of $f$ by one, i.e., $n + 1$ neurons suffice for universal approximation.

### ACKNOWLEDGMENTS

The work of the first author was supported the CONIX research center, one of six centers in JUMP, a Semiconductor Research Corporation (SRC) program sponsored by DARPA. The work of the second author was supported by the Alexander von Humboldt Foundation, and the Natural Sciences and Engineering Research Council of Canada. The authors wish to thank Professor Eduardo Sontag (Northeastern University) for insightful comments on an earlier version of this manuscript.

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
