# OpenReview forum: "Universal approximation power of deep residual neural networks via nonlinear control theory"
_ICLR.cc/2021/Conference — ICLR 2021 Poster_

### Official Review · AnonReviewer2 · 2020-10-22

**Rating:** 6
**Confidence:** 4

**Review:**

This paper studies the universal approximability of residual networks. The main contribution of the paper is to prove that residual networks of width 2n can approximate any continuous function from $\mathbb R^n$ to $\mathbb R^n$ in arbitrary uniform error, for activation functions satisfying some differential equation.

It was interesting for me that only two values of weights suffice for the universal approximation of residual networks. I think that this result can be used to prove the expressive power of compressed/pruned networks.

However, I think that there exists a critical weakness in the connection with the existing universal approximation results. As the authors mentioned in the related work section, (Kidger & Lyons; 2020) proved that feed-forward networks of width n+m+1 can approximate any continuous function from $\mathbb R^n$ to $\mathbb R^m$ in arbitrary uniform error, for general activation functions. In my opinion, this result can be easily extended to residual networks since residual networks can approximate by any feed-forward network by choosing some large s and small W in Eq. (3.1), i.e., ignore the residual connections. If this approximation is valid, the main contribution given by Theorem 4.4 is significantly weakened.

Due to the weakness I mentioned, my decision tends toward rejection.

------------------------------------------------------------------------------------
after reading the authors' response and revision, I raise my score to 6.

---

### Official Review · AnonReviewer1 · 2020-10-28
**The main result (the universal approximation property) itself is rather natural and weak. Possibly, the technique, used for the proof is a new interesting tool itself, however, it is not clear. Authors should somehow emphasize their new ideas in the main text.**

**Rating:** 6
**Confidence:** 4

**Review:**

The authors consider residual networks. They interpret a residual network as a discrete analogue of some dynamical control system. Using the results of the control-theoretic problem of controllability they proved the universal approximation property of the deep neural networks.

The universal approximation property is rather weak; it is usually true for wide classes of models and activation functions. For example, in the case of single-layer networks, it is sufficient that the activation function is not a polynomial, see Pinkus, Allan. "Approximation theory of the MLP model in neural networks." Acta numerica 8.1 (1999): 143-195.

In this paper, the universal approximation property is proved for a very specific class of activations (satisfying a certain diffur). It is not clear why this is a significant requirement. The authors mention that it is possible to extend the results to other activation functions, and explain how to do this for ReLU, but what to do with other activations is still unclear. Of course, approximation using a dynamic system is a more complex model than a conventional single-layer network, but this requirement looks exotic.

So, the authors consider a special type of a complex nonlinear model, parametrized by an infinite-dimensional control, for which the natural and rather weak property is proved.

The proof technology itself is quite interesting and is based on the ideas of Lie algebras. But this is all hidden in the Appendix; if a person reads only the main text, all these ideas are not visible, and the results are not particularly impressive themselves.

Authors should somehow emphasize their new ideas in the main text, or at least make it more interesting.

References to literature are quite relevant in the work and the related work section is OK.

As a conclusion I can say that
- the results itself is rather natural and weak. Nowadays for residual networks we have significantly stronger results about their approximation capabilities, see e.g. Dmitry Yarotsky. “Optimal approximation of continuous functions by very deep ReLU networks.” https://arxiv.org/pdf/1802.03620.pdf
- possibly, the technique, used for the proof is a new interesting tool itself, so that it can be used to solve other similar problems. However, the authors did not mention anything about this in the main text of the paper, and for me it is difficult to assess the novelty as I am not the specialist in the control-theoretic problem of controllability, results of Agrachev et al., etc.

Therefore, I think that in the current state the paper can not be accepted for a publication.

=========

After reading authors comments.

- Well, still the Point 1 is generally not true in my opinion. The authors cited the work of Kidger & Lyons "Universal approximation with deep narrow networks", in which universal approximation is proved for a wide class of activation functions and long narrow networks. Moreover, if you do not strive to make the network as narrow as possible, then it is easy to convert a shallow wide network into a deep narrow one (but not vice versa). Point 2 seems quite possible. Point 3 coincides with what I mentioned --- the authors proved UAP for several examples of activations, although really quite important ones.

- "Results will be accessible without the need for the technicalities". Well, I personally do not understand this statement in the conference article. From the main text, the reader only sees that a relatively basic property (universal approximation) is fulfilled for a couple of previously uncovered activation functions (or for a specialized class), and it is not clear why, since possibly new ideas are not in the main text.

- This work clearly has strengths and weaknesses, which, in principle, are more or less clear. On the one hand, the universality property for approximations by dynamical systems and a specific class of activation functions is proved. How important, new and interesting is it? It doesn't seem to be super strong result, because this property is relatively weak, completely expected, and has already been proven many times before in various situations (wide shallow networks, narrow deep networks, dynamic systems with some classes of activation functions). In approximation theory, people have been studying more advanced questions like approximation rates for a long time. I also do not see any significant practical use of the results. On the other hand, it is possible that the authors have developed some new method when proving their results. But in order to check this fact the reviewer should check the whole appendix. If an article is submitted to a journal, the reviewer is expected to read full article, including the Appendix. But for this conference work, as far as I understand, this is not the case, the reviewer is not required to read the Appendix. Accordingly, if the authors could not convince the reviewer of the importance of the work within the main text, then this is their problem. As for the individual points that the authors wrote about - as I have already written, part of what they wrote is quite fair, no one disputes that they have some new results. At the same time, for example, I find it difficult to agree with point 1, as I have already written.

- Well, in principle, there are two ideas that the authors mentioned (about just two values for all weights and memorising data), which are relatively interesting indeed. This is not something that would be mega-important, but it looks like non-obvious facts, which is a plus.

In principle, I can increase the grade for one point.

---

### Official Review · AnonReviewer3 · 2020-10-28
**This paper aims at explaining the universal approximation capabilities of deep residual neural networks through geometric nonlinear control theory. The main contribution is that, the authors provide a general sufficient condition for a residual network to have the power of universal approximation in the sense of L-infinity norm, and it was claimed that the results in this paper is stronger (/more general) than existing literature works in the same topic.**

**Rating:** 6
**Confidence:** 3

**Review:**

Pros:
  [1] The paper is well-written, clear and organized well.
  [2] The approach of relating DNN universal approximation problem to controllability of ensemble of control systems is interesting.
  [3] The results are a bit more general than existing literature in the same topic.


Main concerns:
  The paper contains some interesting ideas and new results, but it seems it is more an incremental work to existing literature and the contribution might be a bit limited and not significant enough. From empirical sense, it has been well observed that such deep residual neural networks has very good approximation capabilities to almost arbitrary functions. From theoretic sense, the existing relevant work by Li et al., 2019 has already established very similar results (also through dynamic system and control theory) for universal approximation capability of such deep residual neural networks, but just in the sense of L-p norm rather than L-infinity norm (where p can be any number between 1 and infinity). Although the authors argue that L-infinity norm is more general than such L-p norm result, the contribution does seems quite incremental to me (especially considering that the p can be any arbitrary large number between 1 and infinity), and I'm not very convinced by the significance of the contribution, in both theoretic sense as well as practical sense. To be clear, I do think that the approach in this paper is somewhat interesting and inspiring and it also contains some valuable and publishable results, but I just not convinced that the contribution is significant enough to be published as a regular paper in such a top conference like ICLR.

-------
After reading the author's responses, I'm ok to promote the rating from 5 to 6.

---

### Official Review · AnonReviewer4 · 2020-10-29
**Nice theory for representation capabilities of ResNets via dynamical systems and Lie Algebra - Fruitful connections**

**Rating:** 7
**Confidence:** 3

**Review:**

The paper gives a fresh perspective to a standard question in neural networks concerning their power to represent or approximately represent continuous functions. Here they focus on  deep residual neural networks where the activation units satisfy certain conditions defined via a differential equation (most common units in practice abide by this property) and present L_\infinity approximation results which is the strongest possible sense of approximating a function.

The paper uses novel ideas and techniques stemming from dynamical systems and the crux of the argument relies on viewing residuals neural nets as a control system that one can influence via the weight assignments. Using the weights, one can try to obtain specific values as the network's output and hence the universal approximation property follows.

Several ideas along those lines have been previously exploited in related work as the authors state. The reviewer believes there is technical novelty in the paper and also the result is quite elegant. Also, highlighting new connections to dynamical systems has recently been a fruitful direction.

Parts of the techniques establish some "expansion" property of the associated control system implying that for any given point in R^d, the dynamical system can be made to pass through it given the appropriate feedback loop. This holds true even for networks with simple
architectures and if the network's parameters take only two distinct values.

The paper overall offers nice theory for an important question although the practical implications seem limited.

Other comments:
Stating the result for functions R^n to R^n isn't it a bit misleading? Isn't it true that once the result is established for f: R^n to R, then it can directly extend to higher dimension as well?

In terms of presentation, there are several ways to improve: it would be better to define separately what the activation units should be (satisfying the differential equations) and then use it in all places where it is needed. Currently, in most statements for Corollaries and Theorems the authors devote 2-3 lines of repeating information which waste space and make the theorems look ugly.

Also, the authors should clarify more what parts of the approach via Lie algebras is novel and how it is compared specifically with prior works. The reviewer (and potentially most people in ML) are not familiar with such techniques and currently it feels it may be an incremental result.

Moreover, the paper uses ideas from dynamical systems to study approximation capabilities and related works missing from the discussion are https://arxiv.org/abs/2003.00777 and https://arxiv.org/abs/1912.04378 , where depth separations are provided. A similar property holds for those networks as the one described above where only two distinct parameters are used for the weights of the network and it would be interesting to see if your results also can be applied to obtain depth separations.

The paper in https://arxiv.org/abs/1602.04485 also offers representation benefits via general activations called semi-algebraic. What is the connection among those gates and your differential equations? A discussion would be appreciated.

Concerning the Lie algebras presentation, some high-level discussion on how exactly it is used without delving into the technical details for non-experts would also be greatly appreciated and help others build on your work. Currently, several parts of the paper read off as a bit mysterious for non-experts.

Minor Typo:
p1: to best of our knowledge

---

### Decision · Program_Chairs · 2021-01-07
**Final Decision**

**Decision:**

Accept (Poster)

**Comment:**

This paper analyzes the universal approximation power of deep residual neural networks based on nonlinear control theory. Some concerns regarding the clarify, significance and connection to practice were raised, and partially addressed by the rebuttal. After discussions, all the reviewers feel positive about the contribution of the paper.